# Rainfall Trends and Malaria Occurrences in Limpopo Province, South Africa

**DOI:** 10.3390/ijerph16245156

**Published:** 2019-12-17

**Authors:** Abiodun Adeola, Katlego Ncongwane, Gbenga Abiodun, Thabo Makgoale, Hannes Rautenbach, Joel Botai, Omolola Adisa, Christina Botai

**Affiliations:** 1South African Weather Service, Private Bag X097, Pretoria 0001, South Africa; katlego.ncongwane@weathersa.co.za (K.N.); thabo.makgoale@weathersa.co.za (T.M.); joel.botai@weathersa.co.za (J.B.); u13387562@tuks.co.za (O.A.); christina.botai@weathersa.co.za (C.B.); 2UP Institute for Sustainable Malaria Control, School for Health Systems and Public Health, University of Pretoria, Pretoria 0002, South Africa; hannes@akademia.ac.za; 3Department of Mathematics, Southern Methodist University, Dallas, TX 75275, USA; gabiodun@smu.edu; 4Faculty of Natural Sciences, Akademia, Centurion 0157, South Africa; 5Department of Geography, Geoinformatics and Meteorology, University of Pretoria, Private Bag X20, Hatfield 0028, South Africa; 6School of Agricultural, Earth and Environmental Sciences, University of KwaZulu-Natal, Westville Campus, Private Bag X54001, Durban 4000, South Africa; 7Department of Information Technology, Central University of Technology, Private Bag X20539, Bloemfontein 9300, South Africa

**Keywords:** malaria, rainfall, time series, trends, annual, seasonal

## Abstract

This contribution aims to investigate the influence of monthly total rainfall variations on malaria transmission in the Limpopo Province. For this purpose, monthly total rainfall was interpolated from daily rainfall data from weather stations. Annual and seasonal trends, as well as cross-correlation analyses, were performed on time series of monthly total rainfall and monthly malaria cases in five districts of Limpopo Province for the period of 1998 to 2017. The time series analysis indicated that an average of 629.5 mm of rainfall was received over the period of study. The rainfall has an annual variation of about 0.46%. Rainfall amount varied within the five districts, with the northeastern part receiving more rainfall. Spearman’s correlation analysis indicated that the total monthly rainfall with one to two months lagged effect is significant in malaria transmission across all the districts. The strongest correlation was noticed in Vhembe (r = 0.54; *p*-value = <0.001), Mopani (r = 0.53; *p*-value = <0.001), Waterberg (r = 0.40; *p*-value =< 0.001), Capricorn (r = 0.37; *p*-value = <0.001) and lowest in Sekhukhune (r = 0.36; *p*-value = <0.001). Seasonally, the results indicated that about 68% variation in malaria cases in summer—December, January, and February (DJF)—can be explained by spring—September, October, and November (SON)—rainfall in Vhembe district. Both annual and seasonal analyses indicated that there is variation in the effect of rainfall on malaria across the districts and it is seasonally dependent. Understanding the dynamics of climatic variables annually and seasonally is essential in providing answers to malaria transmission among other factors, particularly with respect to the abrupt spikes of the disease in the province.

## 1. Introduction

Malaria is a curable life-threatening infectious disease predominantly caused by *Plasmodium falciparum* parasite. It occurs mostly in 87 endemic countries primarily located in the tropics and subtropics zones of the globe [1,2]. Of the 219 million estimated malaria clinical cases worldwide in 2017, 92% or 200 million were reported in sub-Saharan Africa with 93% of malaria deaths [2]. In the past 7 years (2010–2017), malaria disease burden has declined by about 11% globally from 239 million to 219 million cases with incidence rate from 72 to 59 cases per 1000 population at risk globally while incidence rate remained at 219 cases per 1000 population at risk in the Sub-Saharan region [2]. Moreover, within Southern Africa, similar patterns of the disease are noted in specific countries [3], however, the risk remains unabated due to climate change.

In South Africa (SA), about 95% of confirmed malaria cases are due to infections with *Plasmodium falciparum* through the predominant local vector *Anopheles arabiensis* [4]. In SA, malaria is widespread in the low altitude border areas of Limpopo Province, Mpumalanga and the northeastern parts of KwaZulu-Natal [5]. Higher incidences are reported in Limpopo because of its position in the high-risk malaria transmission zone bordering Mozambique, Zimbabwe, and Botswana. Malaria transmission is year-round and almost exclusively due to *P. falciparum* in most of these countries. Transmission of malaria in Limpopo is perennial, peaking during and after warm rainy seasons, typically between September and May [6]. The transmission rate may still vary in and between years depending on the amount of rain, air temperature, and other climatic factors which together may create a conducive environment [7]. 

Efforts have been made to control malaria in SA, which have lead to a significant reduction in morbidity and mortality in the endemic regions [8]. In recent years, there was an abrupt spike in malaria-related morbidity and mortality in the endemic regions of South Africa where malaria cases went from 1377 in 2016 to 30,558 in 2017 and associated death from 17 to 301 in Limpopo Province [9]. The transmission of the disease is known to be associated with many factors such as climate, environment, and socioeconomic conditions, among others. Many studies have shown that there is a significant relationship between climatic conditions and malaria transmission [9,10,11,12,13,14]. Malaria transmission is influenced particularly by rainfall, temperature, and relative humidity. High temperature improves mosquito maturity as well as the frequency of biting and feeding [15]. However, extremely high temperatures can limit the development of the vectors, thereby reducing the transmission of the disease [16]. On the other hand, the abundance and distribution of the vector are influenced by rainfall through the provision or maintenance of breeding sites. However, heavy or excessive rainfall can have adverse effects on the mosquitoes by washing away mosquito larvae [17]. Additionally, the population of egg-laying females, the number of eggs laid, feeding frequency, as well as the survival rate of the mosquito, is influenced by fluctuation in relative humidity [18].

The Limpopo Province remains the largest contributor to malaria transmission among the endemic provinces [19]. The recent surge in malaria-related morbidity and mortality coupled with the continuous high malaria transmission has awakened the urgent need to proffer answers to many unanswered questions that might be responsible for the surge in malaria transmission. The rural remote northeastern area of Limpopo continued to record a high number of malaria cases and has suffered research neglect when compared to other malaria-endemic regions in KwaZulu-Natal and Mpumalanga provinces [7]. Recently, Adeola et al. [9] examined the relationships between malaria cases and maximum, minimum temperatures and rainfall. The study found that although all variables significantly influenced malaria transmission, rainfall is the most significant driver. Therefore, as proposed in the study [9], it is imperative to analyze the characteristics of long-term observed rainfall and its relationship with malaria cases over the entire Limpopo Province. Hence, this study aims at investigating the influence of monthly total rainfall variations on malaria transmission annually and seasonally in the Limpopo Province.

## 2. Materials and Methods

### 2.1. Study Area

Limpopo Province is situated in the northernmost part of South Africa neighboring Mozambique, Botswana, and Zimbabwe. It lies between latitudes 22°2′ S and 25°4′ S and longitudes 26°4′ E and 31°9′ E. Limpopo Province includes five districts: Vhembe, Mopani, Greater Sekhukhune, Capricorn, and Waterberg (Figure 1). The population of Limpopo has grown from 4,995,534 in 2001 to 5,404,868 in 2011 and estimated to be 5,778,400 in 2017 [20]. It occupies an area of about 125,754 km^2^ having the highest altitude of 2126 m. The study area is drain by River Limpopo and its tributaries. Annually, an average temperature of about 24.6 °C is received with a minimum average of 18.9 °C in June, and a maximum average of 28.2 °C in January. In the study area, January is mostly the wettest month with about 420 mm annual rainfall. July is the driest month with only about 2 mm. An average annual relative humidity of about 77.4% is recorded in the study area.

### 2.2. Datasets

In this study, daily malaria cases positively confirmed by through microscopy or Rapid Antigen Detection Tests (RDT) at all primary health care facilities and laboratories (including private clinics and hospitals), routinely verified, collected, and entered into a computerized malaria information system (MIS) by malaria control program officers, were utilized. The daily datasets were aggregated to monthly at the district level. Although imported cases (30,524 and 26%) seems to be significant, for this study, only localized cases of malaria from 1998 to 2017 (87,075 cases and 74% of total malaria cases) were considered. This was done in order to be able to justify its comparison with local climatic variable.

Monthly rainfall data from January 1998 to December 2017 was extracted from the daily district rainfall data provided by the South African Weather Service (SAWS). A daily district rainfall total is calculated as the average of the daily values available in the district. The country was segmented into 94 homogeneous rainfall districts by the SAWS in an attempt to have data coverage of areas with unique rainfall characteristics. Details of the methodology are well described elsewhere by [21]. The coverage and the delineation of the rainfall district are shown in Figure 1A and the locations of the rainfall stations in Limpopo are shown in Figure 1B overlaid on the rainfall district and political/administrative districts of Limpopo. With two or more homogenous rainfall districts making up an entire political/administrative district, the averages of the rainfall districts were taken to derive the monthly total rainfall for the 5 political/administrative districts of Limpopo. Hence, data in rainfall districts 35, 49, and 50 were used for Vhembe, 34, 48, and 49 for Mopani, rainfall districts 48 and 63 for Sekhukhune, 64 and 65 for Capricorn, and lastly 76, 77, 86, and 87 for Waterberg (Figure 1C).

### 2.3. Time Series Analysis

All statistical analyses were executed in R [22]. Monthly total rainfall was taken as the independent variable and monthly malaria cases as dependent variables. The Box–Jenkins approach was utilized to perform time series analysis on the dataset [23]. Trends in rainfall and malaria cases were computed. Both rainfall and malaria cases showing fluctuations across the districts were detrended by transforming the series logarithmically and differencing to induce constant mean and variance. An assumption was that the monthly malaria cases time series data $y_{t}$ and monthly total rainfall $x_{t}$ after their transformation $y^{\prime }{}_{{}_{t}=\log\left(y_{t}+1\right)}$ and $x^{\prime }{}_{{}_{t}=\log\left(x_{t}+1\right)}$ follow a seasonal model of the form $y^{{}^{\prime }}{}_{{}_{t}}=m_{t}+S_{t}+\varepsilon_{t}$ and $x^{{}^{\prime }}{}_{{}_{t}}=m_{t}+S_{t}+\varepsilon_{t}$, respectively; where $m_{t}$ is the mean level in month t; $S_{t}$ is the seasonal effect in month t, and $\varepsilon_{t}$ is the Gaussian random error. The cross-correlation function (CCF) was performed to determine the time lag(s) between transformed monthly malaria case and monthly total rainfall time series. The relationships between rainfall and malaria cases were investigated at annually and seasonally allowing for the characterization of the temporal variability using the multiple coefficients of determination (*r^2^*) analysis. Thus, given a paired of variables $\left(X_{t},Y_{t}\right)$, a linear model $Y=\beta_{0}+\beta_{1}X+e$ can be used to describe the association between the two variables, where *e* is the mean zero error.

### 2.4. Trend Analysis

The trends are computed using parametric and non-parametric tests. The non-parametric Mann–Kendall test is more robust than parametric student’s t-test when the probability distribution is skewed, however, the tests can be used interchangeably, and yielding results that are identical [24]. Hence, the Mann–Kendall test was employed in this study. Both rainfall and malaria time series were aggregated annually and seasonally; (summer = December–February; DJF, autumn = March–May; MAM, winter = June–August; JJA and spring = September–November; SON) in order to determine potential changes at the seasonal scale during 1998–2017. The regional Kendall test (RKT) package in R software was used to compute the trend and breakpoints. The package is developed to compute the Mann–Kendall (MK), the Seasonal and Regional Kendall Tests (SKT and RKT) as well as the Theil–Sen’s slope. The MK, SKT, and RKT are particularly useful to test for consistency in increase or decrease of a trend in a time series based on the Kendall rank correlation. This is also known as monotonic trends [25]. The resulting *p*-values were used to determine if there is a significant difference in monthly, annual, and seasonal rainfall and malaria.

### 2.5. Inter-Annual Variation Analysis

The inter-annual analysis was performed on the transformed annual malaria cases and total annual rainfall to determine if they are correlated. The relative change in malaria cases between successive years was computed using $\delta_{t,k}=\log(Y_{t,k})-\log(Y_{t-1,k})=\log(Y_{t,k}/Y_{t-1,k})$, where $Y_{t,k}$ is the annual malaria case total for year t, and the start month k of the twelve-month period taken as from the 1st of July given that malaria in South Africa is seasonal, starting from the 1st of July the 30th of June of the succeeding year [10,26]. In addition, the relative change in total rainfall for a period of 12 months before malaria with a lag l of one to three months was computed. A first-order auto-correlated (AR1) model was used to regress malaria against rainfall with the correlation coefficients determined.

### 2.6. Wavelet Coherence Analysis

Wavelet coherence analysis was used to investigate the interconnectedness between rainfall and malaria cases over the five districts. The choice of this approach is based on its ability to identify simultaneously, the time intervals and the frequency bands, where two time-series are correlated. Although the technique is based on the logic of Fourier analysis, it addresses the later limitations by using different scales to analyze different frequencies. This is contrary to the Fourier analysis that uses the same scale for all frequencies [27]. As a result, the wavelet transformation can use good frequency resolution and poor time resolution at low frequencies, as well as good time resolution and poor frequency resolution at high frequencies. Furthermore, the approach has been identified as a most efficient method among the various methods developed to study non-stationary data [27,28]. The methodology has also been considered in many other studies related to climatology and epidemiology [29,30]. As defined in Fourier analysis, the univariate wavelet power spectrum can be broadened to analyze statistical relationships between two time-series x(t) and y(t) by computing the wavelet coherence, using the formula: $$R_{x,y}\left(f,\tau \right)=\frac{\left|\;\langle W_{x,y}\left(f,\tau \right)\rangle \;\right|}{|\;\langle W_{x}\left(f,\tau \right)\rangle \;{|}^{1/2}\;.\;|\;\langle W_{y}\left(f,\tau \right)\rangle \;{|}^{1/2}\;},$$
where 〈 〉 denotes smoothing in both time and frequency; $W_{x,}\left(f,\tau \right)$ represents the wavelet transform of series x(t);
$W_{y}\left(f,\tau \right)$ is the wavelet power transform of series y(t); and $W_{x,y}\left(f,\tau \right)=W_{x}\left(f,\tau \right).\;W_{y}\left(f,\tau \right)$ is the cross wavelet power spectrum. The wavelet coherence provides local information about the extent to which two non-stationary signals x(t) and y(t) are linearly correlated at a certain period or frequency. The $R_{x,y}\left(f,\tau \right)$ is equal to 1 when there exists a perfect linear relationship at a particular time and frequency between the two signals [27]. The wavelet coherence MATLAB code by (Torrence and Compo, 1998) was adopted for this study. In the code, the climate variables and mosquito abundance are denoted by x(t) and the number of malaria cases is denoted by y(t). The code is developed to handle the seasonality and temporal autocorrelation of the data.

## 3. Results

For the period under investigation, a total of 117,599 cases of malaria were reported of which 87,075 were localized cases in the five districts of Limpopo Province. In the study area, the male population accounts for 57% of the total malaria cases. A total of 1274 deaths associated with malaria was recorded during the period under investigation with year 2017 accounting for about 24% of the total deaths. A summary of the overall data including the sex ratio is given in Table 1.

Figure 2a–e, shows the time series of monthly malaria cases and total rainfall in each district from 1998 to 2017. The time series analysis showed a few peaks and fluctuations in all the selected districts. There were remarkable peaks in malaria cases in 2000 across all the districts. For instance, Vhembe recorded 767 and 898; Waterberg 24 and 23; Mopani 255 and 352; Capricorn 46 and 42; and Sekhukhune recorded 10 and 8 cases in October and November, respectively, with a noticeable decline afterward. For the period of study, Vhembe recorded the highest total cases of malaria (55,040 or 63%), followed by Mopani with 28,814 (33%), Capricorn 1464 (2%), Waterberg 1069 (1%), and Sekhukhune with 688 (1%) cases. The monthly mean of malaria cases ranged from the least of 2.07 for Sekhukhune to the highest of 198 for the Vhembe district. Similarly, the median ranged from 0 to 113, corresponding to Sekhukhune and Vhembe districts, respectively. In addition, Vhembe recorded the highest standard deviation (230.17) while Sekhukhune recorded the least (6.04). 

Figure 3 and Figure 4 depict the spatio-temporal (monthly) distribution of total rainfall and malaria cases in study area from 1998 to 2017. The figures were plotted using the inverse distance weight (IDW) technique; a simple interpolation method in spatial analysis. Figure 3 indicates that total rainfall amount varies over time and space. The highest rainfall was received in Mopani district, followed by Sekhukhune district. High spatial and temporal variability of rainfall is found in the hot desert climate region of Vhembe as classified by [31]. Additionally, a higher amount of rainfall was received in the month of January with highs and lows of about 118 and 78 mm respectively while the lowest rainfall was recorded in August with highs and lows of about 4 and 1 mm, respectively.

Figure 4 depicts spatial distribution of malaria cases during the study period. The results indicate that the highest malaria cases were observable in January whereas few cases were recorded in the austral winter months of June, July, and August. In addition, malaria transmission was less in June signifying the low period of malaria during the season with a rise in cases from July signifying the start of season across the districts. 

Figure 5 depicts seasonal distribution of rainfall across the districts. It is noted that most of the rainfall was received over Mopani and Vhembe districts with much of the rain received during the summer period DJF. During the spring season (SON) the highest amount of rainfall was received in Sekhukhune district as depicted in Figure 5.

As shown in Figure 6, strong seasonal variation in malaria cases are observed, with seasonal peaks in DJF and low in JJA.

Table 2 shows the trend and significance trend calculated at 95% confidence interval. According to the trend analysis, the annual variability around the mean values of rainfall in the districts is pronounced (about 1061, 808, 1135, 654, and 1035 mm in Vhembe, Waterberg, Mopani, Capricorn, and Sekhukhune, respectively), regardless of the smoothing effect induced by averaging the data. Given the Theil-Sen’s slope, a decrease in the annual average rainfall particularly in Vhembe and Waterberg is apparent.

The rainfall is estimated to have decreased by about 170 mm. The observed negative trends, particularly in Vhembe, Mopani, and Sekhukhune are statistically in-significant at 95th significant level. No trend is observed in Waterberg and Capricorn. Seasonally, all the districts except Capricorn experienced a decreasing rainfall trend that is not statistically significant. In general, the results depict drier condition over the Limpopo Province; this is alluded to by previous study [21].

On the other hand, malaria cases exhibited negative trends in Vhembe, Mopani, and Capricorn whereas no trends were detected in Waterberg and Sekhukhune. The negative trends in malaria are statistically significant except in Mopani (Table 1). Although negative trends are evident in the summer (DJF) and autumn (MAM), the trends are not statistically significant particularly in Vhembe and Mopani.

The Spearman correlation analysis of the monthly malaria cases with total rainfall indicated a correlation of r = 0.54, 0.40, 0.53, 0.37, and 0.36 in Vhembe, Waterberg, Mopani, Capricorn, and Sekhukhune respectively all statistically significant at *p* < 0.05. Inter-seasonal correlation between malaria cases and rainfall was performed. In Vhembe, the result depicted a high correlation between SON rainfall and DJF malaria (r = 0.82, *p* < 0.001) and a correlation of (r = 0.73, *p* < 0.001) between DJF rainfall and MAM malaria. A weak correlation but significant (r = 0.21, *p* < 0.02) was obtained between SON rainfall and MAM malaria while a negative but strong correlation was obtained between JJA malaria and SON rainfall. Overall, years of high malaria cases are detected to be preceded by early rainfall in winter (JJA) leading to an abundance of mosquito population in Spring (SON) and transmission reaching its peak in summer (DJF).

Figure 7a–e further highlight the relationship between rainfall and malaria cases over the five districts, based on the wavelet spectrum analysis. The relationship is noticeably stronger over Vhembe (Figure 7d) and Mopani (Figure 7b) than Capricorn (Figure 7a), Sekhukhune (Figure 7c), and Waterberg (Figure 7e). Moreover, the intensity of the transmission is more prominent over the two districts than the others. For instance, high frequency periodicities with annual cycles are visible almost throughout the years for Vhembe (except for 2002–2003 and 2007) and Mopani (except for 2013–2016). 

The intensity and periodicities are less significant in more years over the other three districts, especially Sekhukhune. Furthermore, the in-phase (right arrows) on the figures indicate that rainfall leads malaria occurrence in all the districts. In other words, malaria transmission over the districts is instigated by previous rainfalls. The simulation results indicate that about 40–55 mm of rainfall received triggered the transmission of malaria across the study area. In addition, cross-correlation function analysis indicated an averaged lagged time of 1–2 months between total rainfall and malaria across the entire districts except for Waterberg where the lag time is between 2–3 months.

## 4. Discussion

Malaria control program in South Africa has significantly helped to reduce the burden of the disease. However, there are occasional surges in the occurrence of the disease. The number of malaria cases increased significantly in 2000 leading to a major outcry in the health sector. Similarly, malaria cases have recently increased considerably from 1377 in 2016 to 30,558 cases in 2017 in the endemic areas of Limpopo thereby raising pertinent questions on the identification of the major driver of the sporadic increase. Using about two decades of surveillance data between 1998 and 2017, this study illustrated the spatiotemporal patterns of malaria cases as well as investigated the effect of rainfall variations on malaria occurrence in the five districts of Limpopo Province of South Africa that has witnessed a renaissance of malaria transmission. The result indicated significant spatial heterogeneity in the distribution of both malaria cases and total rainfall across the different districts. The results also showed that malaria is not prevalent in all the districts but mainly in Vhembe and Mopani districts with pockets of confirmed high malaria cases in Capricorn. The spatial variability in rainfall amount and trends observed across the study area can be attributed to the topography/relief of the study area [10,21]. 

The study area consists of several mountains such as the Drakensberg, Strydpoort, Lebombo, and the Soutpansberg Mountain in the northern region of the study area. The Soutpansberg Mountain particularly is associated with orographic effect causing the eastern region to receive more rainfall, which tend to influence malaria transmission [32]. This condition indicates that the interaction between the complex topography and climatic conditions, cannot be neglected when investigating the spatial variability of rainfall and its corresponding trends in the study region. Furthermore, the spatial variation in malaria cases across the regions might be due to variation in both environmental and climatic suitability for the malaria vector and parasite to survive. This is largely demonstrated by the results of this study, which showed the high variation in the amount of rainfall received across the different regions and the changes therein. Other important factors could be the presence of large commercial farms particularly at Vhembe and Mopani districts that attract many farm workers from across neighboring countries such as Zimbabwe and Mozambique, which are high malaria endemic areas.

The cross-correlation function indicating an averaged lagged time of one to two months between rainfall and malaria means that rainfall can upset malaria transmission in about one month after a significant rainfall and can continue to have impacts up to the second and third month but with a negative effect at lag above five months. The lag time under an optimum condition is an ideal time for the development and completion of the vector-parasite-host cycle. The development of female *Anopheles* mosquito from egg to larva to pupa to adult and to parasite takes between 5–14 days while the second stage involving the development of the *Plasmodium* parasites (gametocyte to sporozoites) takes about 10–18 days; and the final stage which is the incubation period in the human host from infection to malaria symptoms takes about 14 days [15]. Hence, the overall effect of an increase in rainfall is its impact on malaria transmission in that month plus its impact on continuous months.

Although temperature is not considered in the current study, it remains a significant determinant of the transmission of the disease. The *Anopheles* mosquitoes and the *Plasmodium* parasites need ≥16 °C and ≤32 °C of temperature for their development and survival [33]. Similarly, recent study by [34] indicated that optimal malaria transmission is at 25 °C while transmission dramatically decreases at temperatures >28 °C. However, has reported in the previous study, temperatures in the study area does not vary significantly [9]. Hence, it is assumed that the study area receives adequate temperatures through the year for the survival of both the vector and the *Plasmodium* parasites. Therefore, other climatic factors particularly rainfall are triggers for malaria transmission as the length of the aquatic stage of the mosquito life cycle are principally determined by temperature following adequate rainfall [35]. The reasons for the resurgence of malaria cases after the last outbreak in the year 2000 might appear to be complex as rainfall might not solely be the major driver as other factors such as socio-economic activities in the study area are largely not considered in the current study. However, breakpoints detected in the trend analysis further substantiate the high amount of rainfall received in the late spring SON and summer DJF of 1999/2000, 2007/2008, 2010/2011, and 2016/2017. Although socioeconomic development in the study area largely remained unchanged, an increase is noticed in the movement of non-immune migrants for instance travelers and farm workers particularly in Vhembe, Mopani, and Waterberg where imported malaria cases are on the increase annually. Uncontrolled population movement impairs malaria transmission problem in the study area particularly in Waterberg where 71% of malaria cases are imported. Therefore, the investigation of the impact of the movement of the human population can help to improve malaria control and prevention strategies [36].

As indicated in the wavelet analysis Figure 7a–e, the less significant correlation between rainfall and malaria cases in Capricorn, Waterberg, and Sekhukhune districts may be because of other unfavorable conditions influencing malaria transmission in the regions which are not considered in this study. It could also be that malaria control practices (such as indoor residual spraying) are more productive in these districts. A time lag of one to three months has been established between rainfall and malaria incidence in Limpopo Province [37]. Additionally, [9] has shown that monthly mean minimum temperature and monthly total rainfall can account for about 65% variation in malaria transmission at the two-month lagged effect in Mutale, a local municipality in Vhembe district of Limpopo. The results of the seasonal and wavelet analysis suggest that the transmission of malaria can be expected in the summer months when about 40–55 mm of rainfall is received in the spring months (SON) with other favorable conditions fulfilled.

## 5. Conclusions

In this study, we analyzed approximately two decades of temporal record of malaria cases as well as total monthly rainfall in Limpopo Province of South Africa. The study aimed at investigating the annual, inter-annual (seasonal) and monthly variation in malaria cases as accounted for by the changes in rainfall, notwithstanding the inherent biased statistical moments due to the coarse temporal aggregation in monthly time series. The findings suggested that the exogenous environmental factor plays a role in the temporal dynamics of malaria and does so at different temporal scales. Moreover, the seasonal dynamic of the role of rainfall was identified in which the December to February rain season played a major role in influencing malaria cases in the subsequent year. The analysis indicated that there is a lag of one to two months in the relationship between rainfall and malaria. In addition, the results indicated that about 68% in the variation of malaria cases in summer (DJF) can be explained by spring (SON) rainfall in Vhembe district. Hence, malaria control measures such as larviciding and the use of indoor residual spraying should be embarked upon when about 40–55 mm of rainfall is been received in the spring month. However, this is to be applied with caution as other factors such as soil type, evaporation, infiltration, and soil moisture saturation are known to play a major role for the vector breeding habitats [38,39]. 

Although rainfall among other climatic factors (temperature and humidity held constant) is the major driver of malaria, there is a need to investigate other environmental and socioeconomic drivers. Malaria is termed a disease of poverty [40] meaning that the disease is more associated with low-income groups with poor housing and sanitation conditions, poor access to health services, among other socioeconomic factors. Such factors are currently not available as datasets. Hence, further studies are recommended to determine the impact of these potential factors on the transmission of malaria in the province in an effort to develop a robust early warning system.

A limitation of this study is the few amounts of weather stations located in the region. This is particularly so because of the remote nature of many endemic parts of the study area. The topography of the area has a great impact on the climate distribution and hence a dense network of weather stations might improve the observed results. This study will be coupled with future studies aimed at investigating other malaria transmission factors such as population movement, immunity, socio-=economic status (housing condition, electricity, sanitation, etc.), and vector control measures in an effort to developing a robust malaria early warning system in southern Africa.

## Figures and Tables

**Figure 1 ijerph-16-05156-f001:**
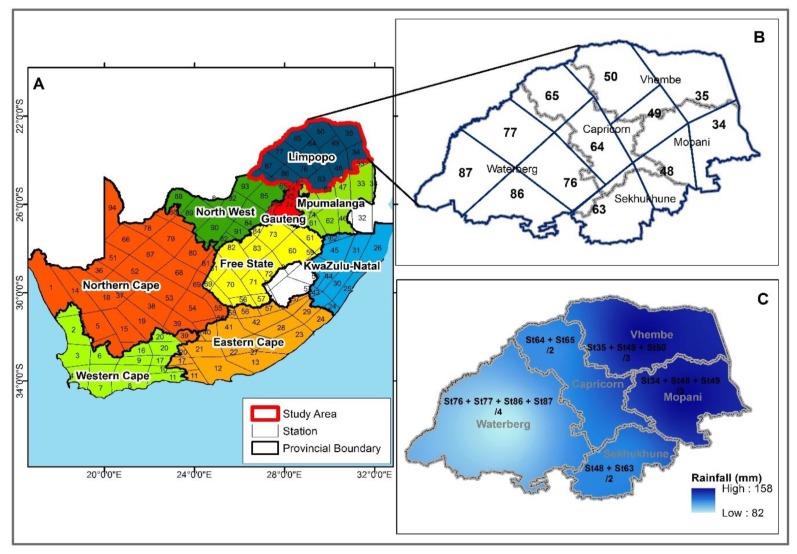
Map of South Africa showing: (**A**) Rainfall districts of South Africa with provincial boundaries, (**B**) Limpopo rainfall districts, and (**C**) spatial distribution of aggregated mean total rainfall across the Limpopo Province in December 2016.

**Figure 2 ijerph-16-05156-f002:**
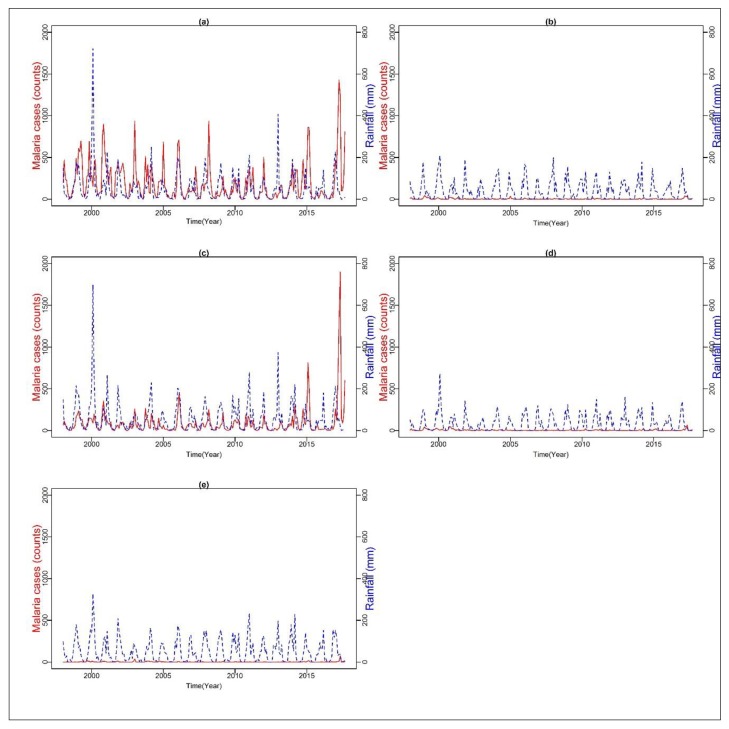
Time series of monthly malaria cases count on left Y-axis (Redline) and rainfall mm on right Y-axis (Blueline) over (**a**) Vhembe, (**b**) Waterberg, (**c**) Mopani, (**d**) Capricorn, and (**e**) Sekhukhune from 1998–2017.

**Figure 3 ijerph-16-05156-f003:**
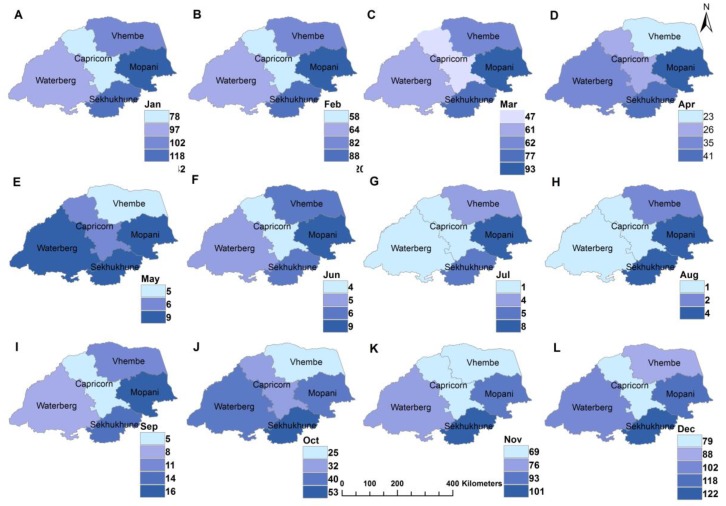
Spatial-temporal distribution of monthly mean total rainfall in Limpopo Province, 1998–2017, **A**–**L** representing January to December respectively.

**Figure 4 ijerph-16-05156-f004:**
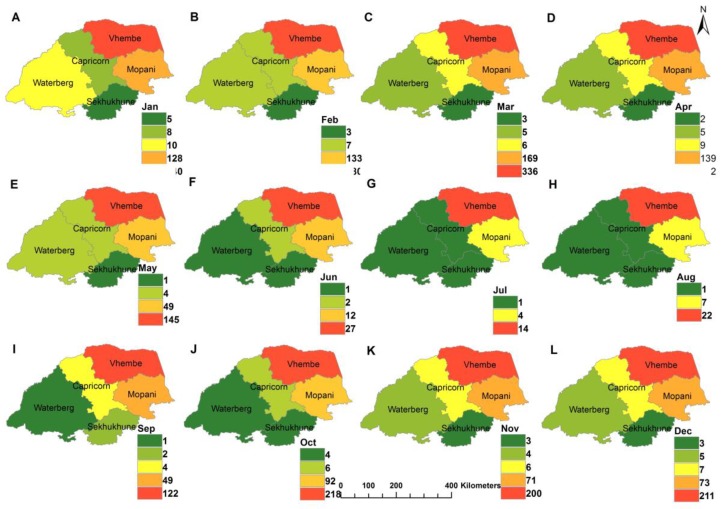
Spatial-temporal distribution of monthly malaria cases in Limpopo Province, 1998–2017, **A**–**L** representing January to December respectively.

**Figure 5 ijerph-16-05156-f005:**
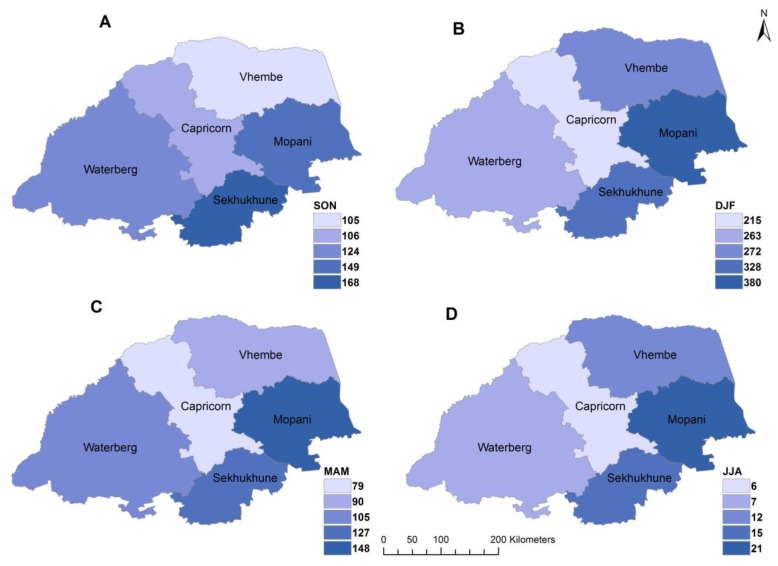
Seasonal time series of total rainfall across districts for (**A**) September–November, (**B**) December–February, (**C**) March–May, (**D**) June–July 1998–2017.

**Figure 6 ijerph-16-05156-f006:**
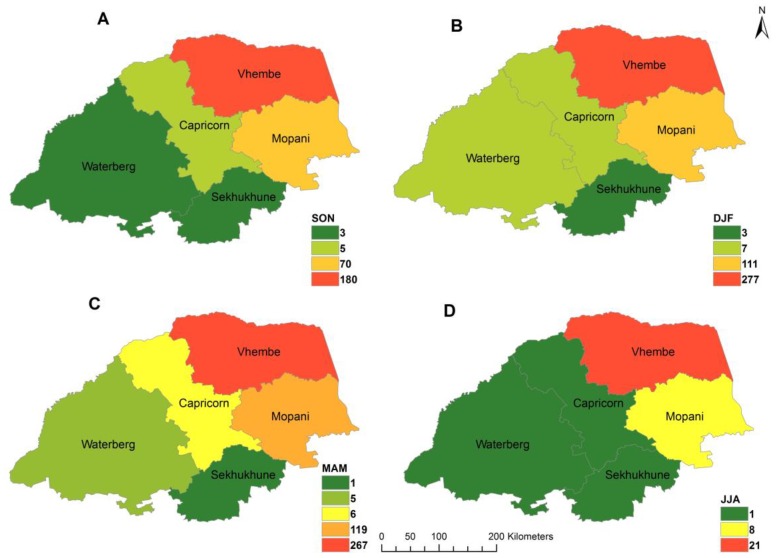
Seasonal time series of malaria cases across districts for (**A**) September–November, (**B**) December–February, (**C**) March–May, (**D**) June–July.

**Figure 7 ijerph-16-05156-f007:**
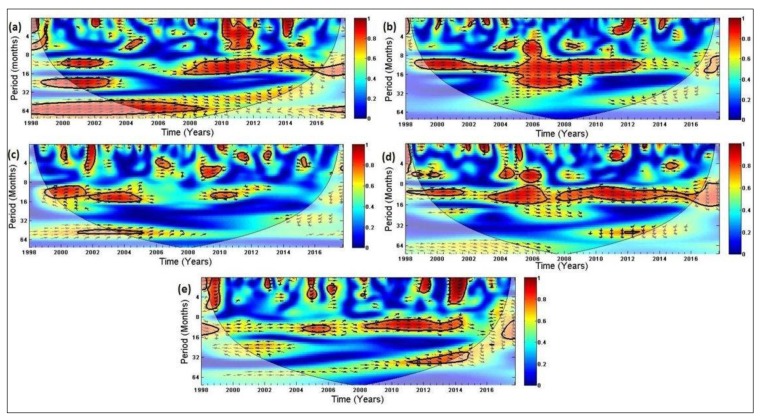
Wavelet cross-coherence of rainfall and malaria cases of (**a**) Capricorn, (**b**) Mopani, (**c**) Sekhukhune, (**d**) Vhembe, and (**e**) Waterberg district from 1998–2017. The arrows indicate the relative phasing of the variables, while the faded regions represent the cone of influence and are not considered for the analyses.

**Table 1 ijerph-16-05156-t001:** Summary of total malaria cases recorded in Limpopo Province from 1998–2017 showing the number and percentage contribution of local and imported cases, sex and associated death.

Year	Malaria Cases	Local(%)	Imported(%)	Female	Male	Death
1998	4082	2961(72.5)	1121(27.5)	1834	2248	59
1999	9093	6154(67.7)	2939(32.3)	4095	4998	112
2000	8553	5658(66.2)	2895(33.8)	4020	4533	71
2001	6215	4049(65.1)	2166(34.9)	2838	3377	50
2002	4372	2982(68.2)	1390(31.8)	1850	2522	40
2003	6094	4387(72.0)	1707(28.0)	2831	3263	87
2004	4650	2875(61.8)	1775(38.2)	2026	2624	54
2005	3003	1673(55.7)	1330(44.3)	1307	1696	26
2006	5867	3734(63.6)	2133(36.4)	2537	3330	51
2007	2885	2114(73.3)	771(26.7)	1188	1697	34
2008	4397	3007(68.4)	1390(31.6)	1808	2589	29
2009	3153	1863(59.1)	1290(40.9)	1318	1835	35
2010	4300	2506(58.3)	1794(41.7)	1756	2544	43
2011	3492	2023(57.9)	1469(42.1)	1296	2196	32
2012	2016	1435(71.2)	581(28.8)	748	1268	22
2013	2408	1563(64.9)	845(35.1)	976	1432	27
2014	5727	4115(71.9)	1612(28.1)	2358	3369	101
2015	5357	4543(84.8)	814(15.2)	2302	3055	83
2016	1377	917(66.6)	460(33.4)	549	828	17
2017	30,558	28,516(93.3)	2042(6.7)	12,975	17,583	301
**TOTAL**	**117,599**	**87,075(74)**	**30,524(26)**	**50,612(43)**	**66,987(57)**	**1274**

**Table 2 ijerph-16-05156-t002:** Trends and *p*-value of monthly malaria cases and total rainfall 1998–2017.

District	Malaria		Rainfall	
	*p*-Value	Trend	*p*-Value	Trend
Vhembe	0.0025	−0.34	0.19	−0.027
Waterberg	0.021	0	0.72	0
Mopani	0.60	−0.019	0.28	−0.025
Capricorn	0.00043	−0.0055	0.91	0
Sekhukhune	0.70	0	0.71	−0.0049

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
