# Peer review of "Rainfall Trends and Malaria Occurrences in Limpopo Province, South Africa"

_ijerph, 2019, doi:10.3390/ijerph16245156_

Round 1

Reviewer 1 Report

Please see attached pdf file.

Author Response

We thank the reviewer for the valuable comments and suggestions on our manuscript. We have addressed these items and we are happy to send you a much improved manuscript. Please see our responses in track changes. Also, the manuscript has been edited by a native English speaking person.

Reviewer 2 Report

The authors describe the relation between Rainfall trends and malaria occurrences in Limpopo Province.

Although aggregated and observational, the paper reports data on human diseases. It is not clear how do the authors faced the ethical issue. Is a there an ethics committee authorization? At least a communication to the committee should be provided.

Although the results refer to a local experience, the association between climatic conditions and spreading of infectious diseases is currently studied in public health. 

The manuscript is interesting but I suggest discussing the results also in a wider context of relation between climatic factors and ID.

https://www.sciencedirect.com/science/article/pii/S0013935119305183?via%3Dihub

Author Response

(The authors gave the same response as above.)

Reviewer 3 Report

The manuscript deals with an important issue that rainfall variations are associated with malaria transmission. However, a few limitations can be addressed to improve the quality of the paper. 

Point 1: The paper uses abbreviations without showing what they mean. For example, what are DJF and SON in line 31?

Point 2: In lines 59-60, please show the actual statistics of malaria-related morbidity and mortality in South Africa.

Point 3: Please describe why the authors chose rainfall as the primary risk factor for malaria transmission, rather than temperature and relative humidity, in line 80.

Point 4: Please describe the method of how the monthly district rainfall was interpolated from daily rainfall observations. (i.e.) Used average daily rainfall observations to calculate the monthly average. Additionally, please describe how the authors treated missing data if there were any missing data in daily rainfall observations.

Point 5: Between lines 116 and 119, data from the monitoring stations 49 and 48 were used twice to estimate monthly rainfall for Bhembe, Mopani and Sekhukhune. Would it be possible that it is the reason the estimations of monthly rainfall were high in those regions?

Point 6: Showing plots for monthly malaria cases and monthly rainfall would be beneficial for readers.

Point 7: The authors log-transformed rainfall and malaria cases. However, it was not clear that they used the transformed rainfall and malaria cases in annual and seasonal level analyses. 

Point 8: It would be easier for readers to detect fluctuations and correlation if the results were shown in tables (lines 170-173, lines 220-225).

Point 9: Limitations of the study can be included in the discussion section.

Author Response

(The authors gave the same response as above.)

Round 2

Reviewer 3 Report

1. Starting from line 139, the authors described they performed CCF to determine the time lag(s). However, it is unclear that the results of CCF analysis were shown in the result section.

2. In Figure 3, the highest amount of rainfall seemed 118 according to the legend whereas the authors stated that the highest was 142 mm in line 222.

Author Response

Reviewer's comment: 1. Starting from line 139, the authors described they performed CCF to determine the time lag(s). However, it is unclear that the results of CCF analysis were shown in the result section.

Author's Response: The results of the CCF is indicated in lines 288-290

Reviewer's comment: 2. In Figure 3, the highest amount of rainfall seemed 118 according to the legend whereas the authors stated that the highest was 142 mm in line 222.

Author's Response: the amount of rainfall has rightly been changed to 118 mm as shown in the figure. An error of omission must have been committed. 
